# Aerobic exercise up-regulates Klotho to improve renal fibrosis associated with aging and its mechanism

**Jinfeng Zhao[1], Yue Guan[1], Yingxiu Jia[1], Yinghua Chen[1], Yue Cai[2]***

**1** Research Center for Human Movement Science, School of Physical Education, Shanxi University, Taiyuan, Shanxi, China, **2** Department of Anesthesiology, Shanxi Province Cancer Hospital/Shanxi Hospital Affiliated to Cancer Hospital, Chinese Academy of Medical Sciences/Cancer Hospital Affiliated to Shanxi Medical University, Taiyuan, Shanxi, China

* 13111081922@163.com

**Data Availability Statement:** All relevant data are within the manuscript and its Supporting Information files.

**Funding:** 1. ZJF [Grant number: 31500962] This work was supported by the National Natural Youth

## Abstract

Renal fibrosis is a major cause of renal dysfunction and is a common pathological event in almost all forms of chronic kidney disease (CKD). Currently, the pathomechanisms of renal fibrosis are not well understood. However, researchers have demonstrated that aerobic exercise can improve renal fibrosis. Klotho is considered to be a negative regulator of renal fibrosis. In this study, we aimed to investigate the role and mechanism of Klotho in the improvement of renal fibrosis through aerobic exercise. We performed a 12-week aerobic exercise intervention in 19-month-old male C57BL/6J mice. Physiological and biochemical indexes were performed to assess renal function and renal fibrosis. The roles of Klotho were further confirmed through knockdown of Klotho by small interfering RNA (siRNA) in C57BL/6J mice.Q-PCR and Western blot were performed to quantify determine the expression of relevant genes and proteins in the kidney. Results: Aging decreased Klotho expression via activated the upstream TGF-β1/p53/miR34a signaling pathway and affected its downstream signaling pathways, ultimately leading to renal fibrosis. Exposure to aerobic exercise for 12 weeks significantly improved renal fibrosis and alleviated the intrarenal genetic alterations induced by aging. Conclusion: Our results showed that aerobic exercise increased Klotho expression by inhibiting the TGF-β1/p53/miR34a signaling pathway and further inhibited its downstream TGF-β1/smad3 and β-linker protein signaling pathways. These results provide a theoretical basis supporting the feasibility of exercise in the prevention and treatment of CKD.

## Introduction

Renal fibrosis is characterized by excessive deposition of extracellular matrix (ECM) which triggers glomerulosclerosis and tubular atrophy, serving as a common pathway for end-stage renal failure in various chronic kidney diseases (CKD) [1]. CKD has emerged as a global health concern, posing a significant threat to human health and exhibiting a sharp increase among the elderly population annually. The estimated prevalence of CKD among elderly patients ranges from 23.4% to 35.8%, which is significantly higher than that of younger cohorts [2].

Science Foundation of China. Website: https://grants.nsfc.gov.cn/pmpweb/login 2. ZJF [Grant number: 20210302124022 ] Basic Research Program of Shanxi Province (Free Exploration Category) website: https://kjt.shanxi.gov.cn/ 3. CY [Grant number: 202103021224428] Basic Research Program of Shanxi Province (Free Exploration Category) website: https://kjt.shanxi.gov.cn/ the funders play role in the study design, data collection and analysis, decision to publish, and preparation of the manuscript.

**Competing interests:** The authors have declared that no competing interests exist.

However, to date, there are no effective therapeutic interventions to effectively intervene in CKD and halt the progression of renal fibrosis. Therefore, it is imperative to investigate the pathogenesis of renal fibrosis to facilitate the development of novel strategies to delay the progression and improve renal function. Klotho, an anti-aging protein predominantly expressed in the kidney, particularly within renal tubular epithelial cells [3, 4], is recognized as a negative regulator of renal fibrosis factors. Clinical investigations have highlighted a significant reduction in Klotho expression in renal fibrosis, indicating an association between Klotho deficiency and the pathogenesis of renal fibrosis [5]. Furthermore, Klotho exerts its influence on renal fibrosis via downstream signaling pathways, including but not limited to the transforming growth factor-β1 (TGF-β1)/Smad3 and wnt/β-catenin pathways [6, 7]. MiR34a, a small non-coding RNA [8], is intricately linked with the progression of kidney diseases, encompassing renal fibrosis, acute kidney injury, and renal cell carcinoma [9–11]. Compelling evidence suggests that p53 is an upstream target of miR-34a, occupying a highly conserved consensus binding site proximal to the miR-34a transcriptional start site, that significantly induces miR-34a transcription [12, 13]. Particularly noteworthy is the pivotal role of TGF-β1 in driving most of the fibrotic processes observed in chronic kidney disease (CKD), where its overexpression precipitates renal fibrosis [6]. TGF-β1 modulates p53 activity by promoting phosphorylation and acetylation of p53 at serine 15 and 9, respectively [14]. Furthermore, miR-34a exerts its pro-fibrotic effects by downregulating the expression of the negative regulator of fibrosis, Klotho, through direct binding to its 3'UTR, thereby fostering renal fibrosis [15]. A study by Liu *et al* demonstrated that dihydromyricetin ameliorated renal fibrosis by modulating the TGF-β1/p53/mir34a/Klotho signaling cascade [9].

Exercise has been demonstrated to mitigate kidney injury and ameliorate renal dysfunction, renal fibrosis, and chronic renal failure [16–19]. Duan et al., observed that an 8-week swimming regimen in spontaneously hypertensive rats reduced renal fibrosis associated with hypertensive nephropathy by modulating the TGF-β1/Smad signaling pathway [20]. Similarly, Bao *et al*., applied incremental load training in aged mice, and found that exercise effectively reduced renal fibrosis in this demographic by modulating the TGF-β1/TAK1/MMK3/p38AMPK signaling pathway [21]. Notably, investigations into the role of klotho in mediating exercise-induced improvements in renal fibrosis remain conspicuously absent from the literature. To explore the link between aerobic exercise and Klotho in a mouse model of renal fibrosis induced by aging mice. We hypothesize that aerobic exercise, in addition to being an impressive Klotho upregulator, inhibit the TGF-β1/Smad3 and Wnt/β-linker protein signaling pathways, forming a delayed benign feedback loop with TGF-β1/p53/miR-34a in renal fibrosis. Hence, this study exposed aged mice to 12 weeks of treadmill running training to assess whether aerobic exercise could ameliorate renal function and fibrosis in aged mice, while also elucidating the underlying mechanisms. Our results are expected to provide novel insights into the prevention and mitigation of age-related renal fibrosis.

## Materials and methods

### Ethical approval

42 SPF male C57BL/6J mice (19 and 2 months of age) were purchased from Beijing Veitonglihua Laboratory Animal Limited Technology Company (Beijing, China). Animal license number: SCXK (Beijing) 2016–0006. The Experimental Animal Welfare Ethics Review Committee of the Scientific Research Institute of Shanxi Cancer Hospital and the China Institute of Radiation Protection (CIRP) approved this study with approval number: CIRP-IACUC-(R)2021008. All experiment procedures were conformed to the Shanxi Cancer Hospita ethical guidelines on the use of animals and the ARRIVE guidelines.

## Animal grouping and exercise training

Mice were given free access to food and water under controlled conditions of temperature (20–26˚C), humidity (40–70%), and a 12-hour light-dark cycle. Mice were divided into four groups: young group (Y, n = 6), young exercise group (YE, n = 6), aging group (A, n = 6), and aging exercise group (AE, n = 6). All mice were adaptively fed for one week, and mice in the YE and AE groups underwent one week of adaptive exercise and 12 weeks of formal aerobic exercise intervention. Briefly, the platform running test was performed on a motorized and speed-controlled platform system (Shanxi, China). Mice were warmed up at a speed of 8 m/min for 5 min, then the speed was increased by 2 m/min every 2 min until 14 m/min with an inclination of 6% for 30 min, and finally, the exercise was ended with a cool-down of 5 m/min for 2 min (S1 File). Mice were constantly monitored while on the treadmill. Mice were immediately removed from the platform if they developed an injury during training. At the end of the experiment, mice (6 per group) were executed, and kidneys and serum were taken for various analyses.

## Klotho interferes with mouse construction

To effectively knock down Klotho in the kidney, we used a small interfering RNA (siRNA)-mediated RNA interference technique. The siRNA targeting 5′-GCGACTACCCAGAGAGAG TAT-3′ in the mouse Klotho gene was synthesized by Shanghai Sangong Bioengineering Company. The scrambled RNA 5′-CGUACGCGGAAUACUUCGATT-3′ was used as the control. Klotho-interfering mice were backgrounded with C57BL/6J mice (2 months old), and after 1 week of acclimatization feeding, the mice were randomly divided into the siRNA-control group (siRNA-control,n = 6), the Klotho knockdown group (siRNA-Klotho,n = 6), Klotho knockdown+exercise group (siRNA-Klotho+Ex,n = 6). Injections were performed with reference to existing studies [22]. SiRNA targeting the Klotho gene (10 nm in 200 μl phosphate-buffered saline) was injected via tail vein for the siRNA-Klotho and siRNA-Klotho+Ex groups of mice, and negative control siRNA was injected for the siRNA-control group, once a week for a total of 8 weeks. During the injection period, aerobic exercise intervention was performed for 8 weeks for the siRNA-Klotho+Ex group. Aerobic exercise program consistent with the previous program. The siRNA-control and siRNA-Klotho groups were not exercised and fed normally. All mice consumed a standard diet during the entire experimental period.

## Tissue collection

At the end of the last treadmill training session, mice were fasted for 12 hours. Mice were executed using the spinal dislocation method and their blood was preserved in EP tubes. The mouse blood samples were centrifuged (3500 rpm, 4˚C) for 10 minutes, and the supernatant was collected and stored at -20˚C. Mice kidneys were immediately removed and washed with cold phosphate-buffered saline, then the kidney tissue was weighed and cut into two parts: one part was stored in fixative and the other part was frozen in liquid nitrogen and stored at -80˚C until extraction.

## Masson's staining

After fixed kidney tissue was dehydrated, made transparent, and wax-impregnated, it was then taken for paraffin embedding and cut into 3- to 4-μm slices. Dewaxing of 4-μm-thick kidney tissue sections was performed through a series of steps. Nuclei were stained with Weigert iron hematoxylin stain (Sigma, H9627) for 5 min, rinsed for several minutes, divided in alcohol hydrochloride for 3 seconds, and rinsed in running water for 5 min. The nuclei were then stained with Lichon red acidic magenta stain (Sigma, F8129) for 3 min, rinsed for several

minutes, fractionated with the phosphomolybdic acid solution for 5 min, and immediately restained with aniline blue stain for 5 min. After a series of rinses with glacial acetic acid, they were dehydrated and made transparent, sealed with neutral adhesive, then finally observed with a microscope (Olympus BX53 Biomicroscope). The percentage of fibrosis was determined as the blue area divided by the area of the entire field of view.

## HE staining

Fixed kidney tissues were taken paraffin-embedded by dehydrated, made transparent and wax-impregnated followed by cutting into 3- to 4-μm slices. The 4-μm-thick kidney tissue sections were dewaxed through a series of steps. The sections were stained with Mayer's hematoxylin (Sigma, H9627) stain for 5 min, washed with tap water immersion, and returned to blue with PBS. Then, the sections were stained with water-soluble eosin stain (Sinopharm, 71014544) for 5 min. Finally, they were dehydrated and made transparent, sealed with neutral glue, and observed with a microscope (Olympus BX53 Biomicroscope).

## Blood analysis

The levels of urea nitrogen (BUN) and creatinine (Cr) in the serum samples of each group of mice were measured by enzyme marker. All operations were performed in strict accordance with the instructions of the kit (Wuhan Elite Biotechnology Co., Ltd.).

## RNA extraction and quantitative PCR

RNA was extracted by Trizol method, 0.1g of kidney tissue was taken and added to buffer at a weight-to-volume ratio of 1:10. The dried RNA was precipitated for 10 minutes and then dissolved the precipitate. 2μl of dissolved RNA was taken and measured by microscopic spectrophotometer for OD260, OD280, and OD260/OD280 values to calculate the purity and concentration of RNA. Quantitative PCR (Q-PCR) was performed on a Light Cycler 480 system (Roche, Switzerland) via TB Green premix Ex Taq II mix (TaKaRa, Dalian, China). Using $2^{-\triangle\triangle CT}$ to calculate the Intra-renal levels of mRNA. The primer sequences are listed in S1 File. The expression level of each gene was normalized to GAPDH.

## Western blot

Kidney tissue was lysed in RIPA buffer, and then the total protein concentration was measured by a BCA protein assay kit (Beyotime, Shanghai, China). The equal amounts of Proteins (40μg) were separated by SDS-polyacrylamide gel electrophoresis (SDS-PAGE) and transferred onto PVDF membranes (Millipore, Billerica, USA). The membranes were blocked with 5% nonfat milk for 1 hour at room temperature. And they were incubated overnight at 4°C with 1:1000-fold diluted specific primary antibodies anti-a-SMA, anti-Kim-1, anti-E-cadherin, anti-Fibronectin, anti-klotho, anti-TGF-β1, anti-P-p53, anti-Wnt1, anti-β-catenin, anti-MMP7, anti-P-smad3. β-actin was used as an endogenous control. After washing, the membranes were incubated with the appropriate horseradish peroxidase-conjugated (HRP-conjugated) secondary antibodies (1:5000) (Boster Biotech, Wuhan, China) for 1 h, with the signals detected by an ECL detection kit (Applygen Technologies Inc, Beijing, China) and imaged in a chemiluminescence-measuring instrument (ChemiDoc XRS+, Bio-Rad, USA).

## Statistics

All experimental data were expressed as mean ± standard deviation (Mean±SD). Data were analyzed using SPSS software (25.0), two-way ANOVA and Tukey's post hoc test were used

for comparison between four groups after tests of normality and chi-squaredness. The data from the si-RNA experiment was analyzed using one-way ANOVA followed by Tukey's post hoc test. At least three independent experiments were performed. Data were significantly different when $P<0.05$.

## Results

### Aerobic exercise improved renal impairment and renal fibrosis in aged mice

To explore the impact of aerobic exercise on renal fibrosis in aged mice. 19-month-old mice were exposed to aerobic exercise for 12 weeks. The results showed that the serum urea nitrogen (BUN) level in group A (14.2±0.61 mmol/L)was significantly higher than that in group Y (10.68±1.6 mmol/L)($P<0.01$), and the serum creatinine (Cr) concentration was significantly higher in group A (70.37±3.69 mmol/L) than that in group Y (61.16±4.37 mmol/L) ($P<0.01$), and the mRNA expression of the renal injury molecule 1 (Kim-1) was also significantly higher in group A than that in group Y. The levels of these variables were reduced in the AE group compared to the A group ($P<0.05$)(Fig 1A and 1B). This implied that aerobic exercise improved the renal impairment associated with aging. HE and Masson staining results revealed severe renal tissue damage in group A, characterized by extensive vacuolar degeneration, atrophy, and necrosis of numerous renal tubules and tubular epithelial cells, along with substantial collagen deposition, compared to mice in groups Y and YE. ($P<0.01$) (Fig 1C). Moreover, the expression level of the tubular epithelial cell marker E-calmodulin (E-cadherin) was significantly decreased in the renal tissue of group A ($P<0.05$), and that of the renal interstitial markers a-SMA and fibronectin was significantly increased ($P<0.01$), indicating that aging was accompanied by renal fibrosis. Notably, the aging-associated changes were improved in the AE group following the 12 weeks of aerobic exercise intervention ($P<0.05$) (Fig 1D and 1E). These data suggested that aerobic exercise contributed to the improvement of renal fibrosis associated with aging.

### Aerobic exercise upregulated Klotho expression in the kidney of aged mice

Since Klotho was an essential endogenous inhibitor of renal fibrosis, we tended to evaluate the effect of aerobic exercise on klotho in mice with aging-induced renal fibrosis. It was observed that Klotho was significantly higher in the YE group than in the Y group ($P<0.05$), suggesting that running exercise significantly upregulated the expression of Klotho. The expression of Klotho was significantly lower in the kidney tissue of group A compared to groups Y and YE, but its expression was increased after 12 weeks of aerobic exercise ($P<0.05$)(Fig 2A, 2B). The above data suggested that aerobic exercise ameliorated renal fibrosis associated with aging by modulating Klotho expression.

### Aerobic exercise promoted Klotho expression in aged mice by inhibiting the TGF-β1/p53/ mir34a signaling pathway

Studies have shown that miR34a binds to the 3'UTR of Klotho to negatively regulate its expression. Therefore, we aimed to investigate whether aerobic exercise can inhibit miR-34a expression in aged mice. Data shown in Fig 3A suggested that aerobic exercise significantly inhibited the expression of miR-34a ($P<0.05$). In addition, we also explored the effect of aerobic exercise on the expression of p53 which acts upstream of miR-34a. The results showed that it was significantly downregulated in the AE group than in group A ($P<0.05$)(Fig 3B). Further analysis uncovered that aerobic aerobic exercise on TGF-β1 expression, as shown in Fig 3C. Aerobic

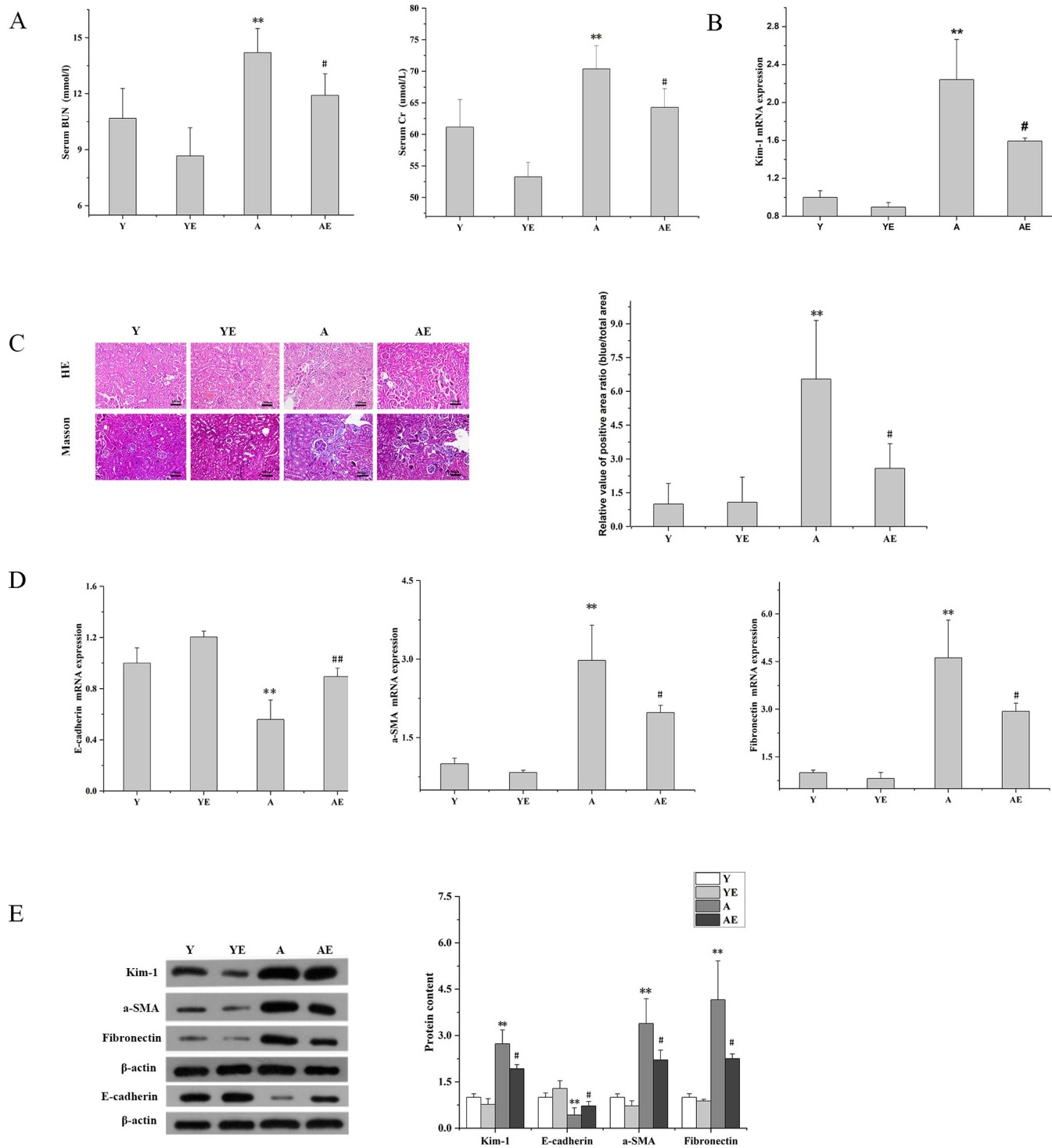

**Fig 1. Aerobic exercise improved renal impairment and renal fibrosis in aged mice.** (A) Serum levels of BUN and Cr were assessed. (B) The mRNA expression of Kim-1 in the kidneys of each group of mice was determined by Q-PCR. (C) Representative micrographs (magnification, x200) of H&E and Masson stained kidney sections from each group of mice, with quantitative results showing the extent of fibrosis between groups. (D) The mRNA expression of renal E-cadherin, a-SMA and Fibronectin in each group of mice was determined by Q-PCR. (E) Western blot analysis (left) and quantification (right) of intrarenal protein levels of Kim-1, E-cadherin, a-SMA, and Fibronectin. *$P<0.05$ vs. Y; #$P<0.05$ vs. A; **$P<0.01$ vs. Y; ## $P<0.01$ vs. A. Data are expressed as mean±standard error (n = 6). Abbreviations: A.U. Arbitrary units; Y: young group; YE: young exercise group; A: aging group; AE: aging exercise group; BUN: blood urea nitrogen; Cr: creatinine; H&E: hematoxylin-eosin; Masson: Masson trichrome staining; Kim-1: kidney injury molecule; E-cadherin: E-calmodulin; a-SMA: a-smooth muscle actin; Fibronectin: fibronectin.

A

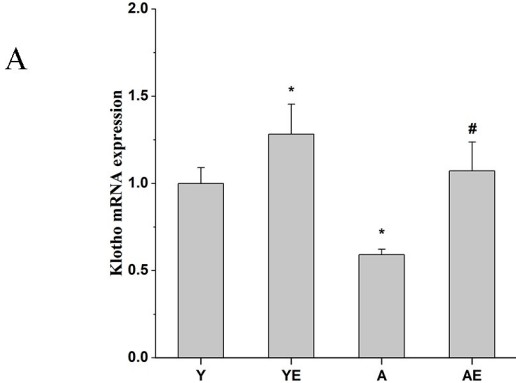

B

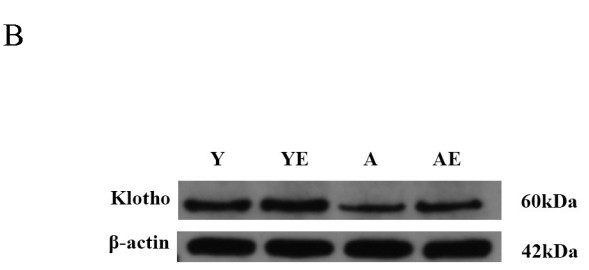

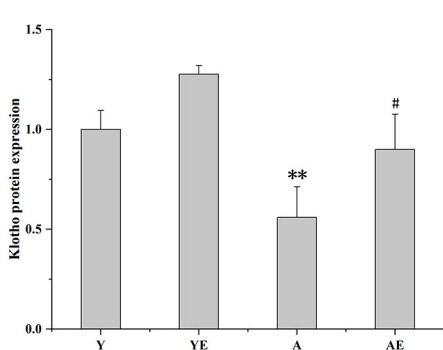

**Fig 2. Aerobic exercise upregulated Klotho expression in the kidney of aged mice.** (A) The mRNA expression of Klotho in the kidney of each group of mice was determined by Q-PCR. (B) Western blot analysis (left) and quantification (right) of intrarenal protein levels of Klotho. $*P<0.05$ vs. Y; $^{\#}P<0.05$ vs. A; $**P<0.01$ vs. Y; $^{\#\#}P<0.01$ vs. A. Data are expressed as mean±standard error (n = 6). Abbreviations: A. U. Arbitrary units; Y: young group; YE: young exercise group; A: aging group; AE: aging exercise group.

exercise significantly inhibited TGF-β1 expression ($P<0.05$) and improved renal fibrosis associated with aging. Thus, aerobic exercise promoted Klotho expression and inhibited renal fibrosis by inhibiting the TGF-β1/p53/ miR34a signaling pathway.

## Aerobic exercise inhibited the expression of Wnt/β-catenin/MMP7 signaling pathway in aged mice

It has been shown that activation of Wnt/β-catenin protein promotes the development of renal fibrosis, and that Klotho is an endogenous antagonist of Wnt. Therefore, we investigated whether aerobic exercise could inhibit the Wnt/β-catenin signaling pathway by activating Klotho. Results showed that the expression levels of Wnt, β-catenin and its downstream MMP7 were significantly elevated in group A compared with groups Y and YE ($P<0.01$), but the expression was reduced after 12 weeks of running table exercise ($P<0.05$). These data suggested that aerobic exercise may improve renal fibrosis by inhibiting Wnt/β-catenin (Fig 4A and 4B).

## Aerobic exercise suppressed the expression of the TGF-β1/Smad3 signaling pathway in aged mice

Subsequently, we investigated whether aerobic exercise altered the TGF-β1/Smad3 signaling pathway in renal tissues. Klotho protein could compete with TGF-β1 for binding, resulting in

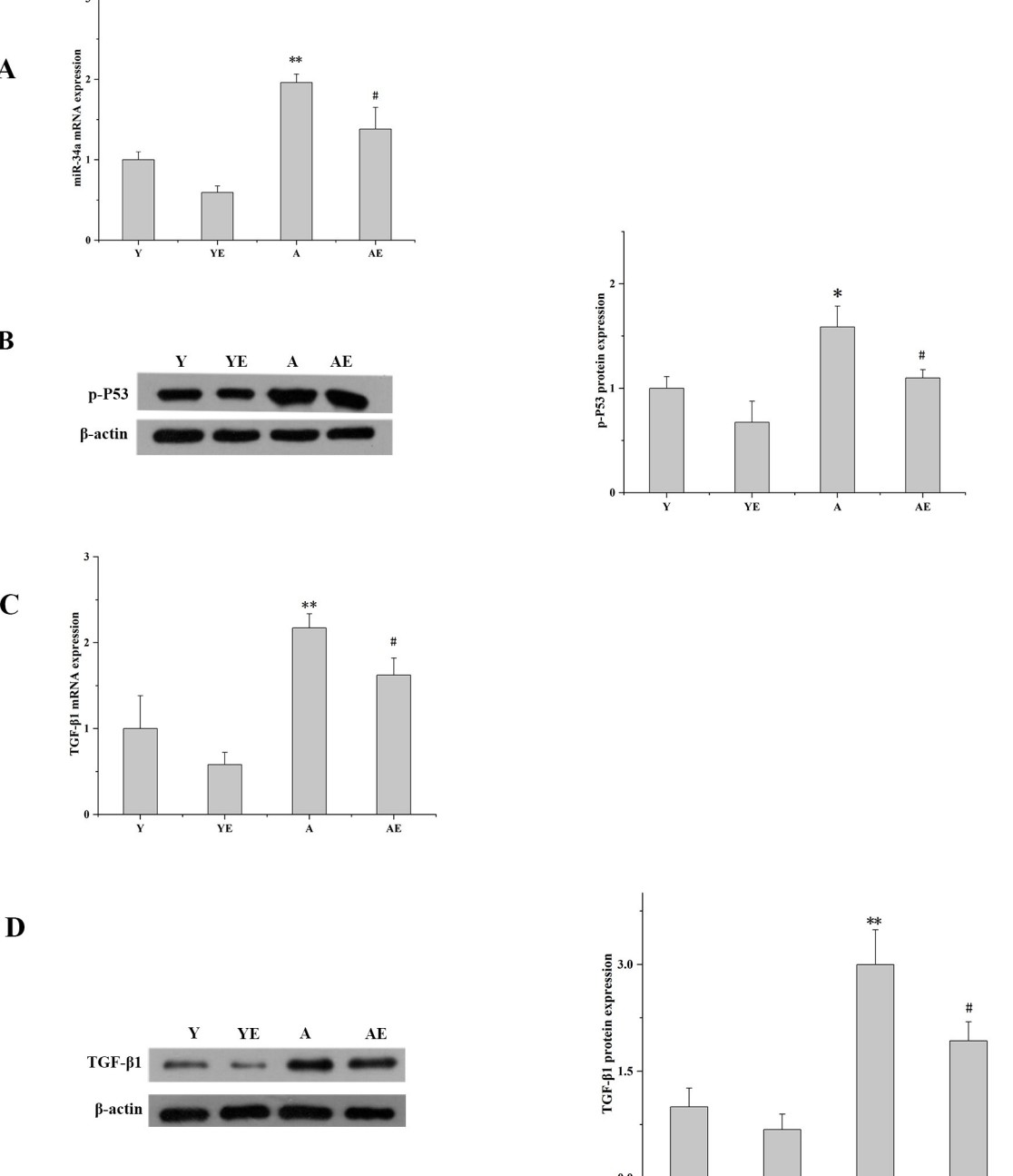

**Fig 3. Aerobic exercise may promoted Klotho upregulation by inhibiting the TGF-β1/p53/ mir34a signaling pathway.** (A) The mRNA expression of miR-34a in the kidney of each group of mice was determined by Q-PCR. (B) Western blot analysis (left) and quantification (right) of intrarenal protein levels of p-P53. (C) The mRNA expression of TGF-β1 in the kidney of each group of mice was determined by Q-PCR. (D) Western blot analysis (left) and quantification (right) of intrarenal protein levels of TGF-β1. $*P<0.05$ vs. Y; $^{\#}P<0.05$ vs. A; $^{**}P<0.01$ vs. Y; $^{\#\#}P<0.01$ vs. A. Data are expressed as mean±standard error (n = 6). Abbreviations: A.U. Arbitrary units; Y: young group; YE: young exercise group; A: aging group; AE: aging exercise group;p-P53:Phosphorylation of P53; TGF-β1:transforming growth factor-beta1.

reduced binding of TGF-β1 to cell surface receptors, which in turn inhibited the TGF-β1 downstream signaling pathway and improved renal fibrosis. It was observed that the expression levels of TGF-β1 and Smad3 in group AE were significantly lower compared to levels in

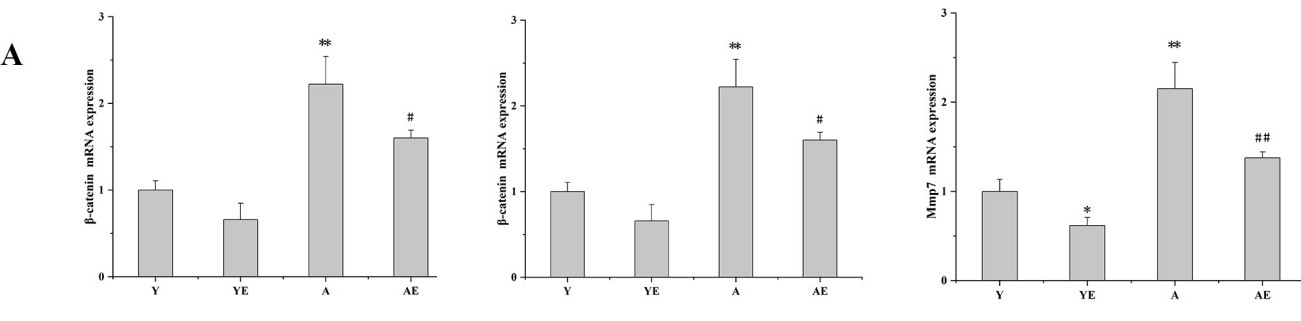

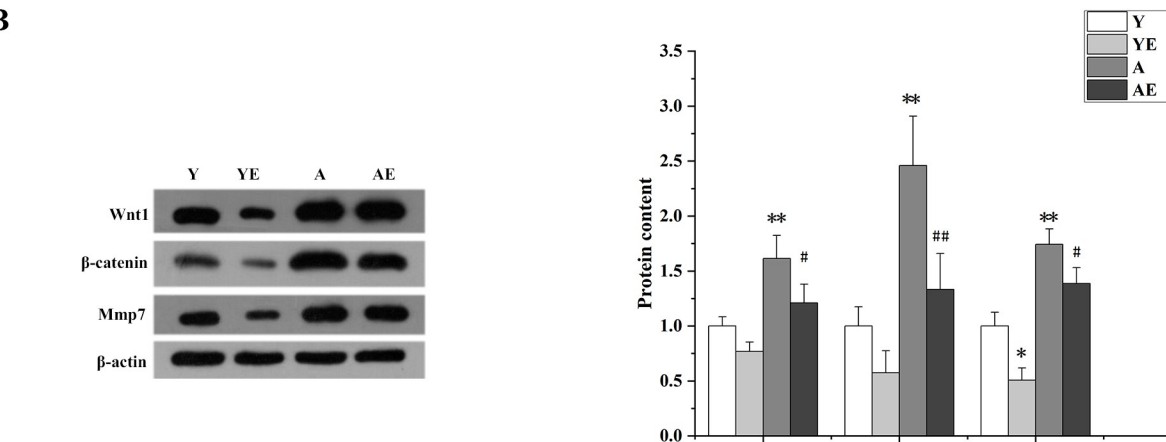

**Fig 4. Aerobic exercise may promoted the expression of Wnt/β-catenin /MMP7 signaling pathway.** (A) The mRNA expression of Wnt1, β-catenin and Mmp7 in the kidney of each group of mice was determined by Q-PCR. (B) Western blot analysis (left) and quantification (right) of intrarenal protein levels of Wnt1, β-catenin and Mmp7. $^*P<0.05$ vs. Y; $^\#P<0.05$ vs. A; $^{**}P<0.01$ vs. Y; $^{\#\#}P<0.01$ vs. A. Data are expressed as mean±standard error (n = 6). Abbreviations: A.U. Arbitrary units; Y: young group; YE: young exercise group; A: aging group; AE: aging exercise group; Mmp7: matrix metalloproteinase-7.

group A ($P<0.05$), which indicated that aerobic exercise significantly improved renal fibrosis (Fig 5A).

## Rescue effect of aerobic exercise in Klotho-inhibited mice

To further elucidate the effect of aerobic exercise on Klotho, C57BL/6J mice were injected for 8 weeks in the tail vein. SiRNA targeting the mouse Klotho gene was injected into the siRNA-Klotho and siRNA-Klotho+Ex groups of mice, and a negative control siRNA was injected into the siRNA-control group. Mice in the siRNA-Klotho+Ex group were continuously exercised for 8 weeks during the injection period.The results show that 8 weeks of Klotho inhibitor significantly knocked down Klotho expression compared to the siRNA-control group. However, in the siRNA-Klotho+Ex group, aerobic exercise led to a notable increase in Klotho expression after 8 weeks of intervention ($P<0.0001$) (Fig 6A). The results shown in Fig 6B indicated that BUN, Cr, and Kim1 were significantly upregulated in the siRNA-Klotho group

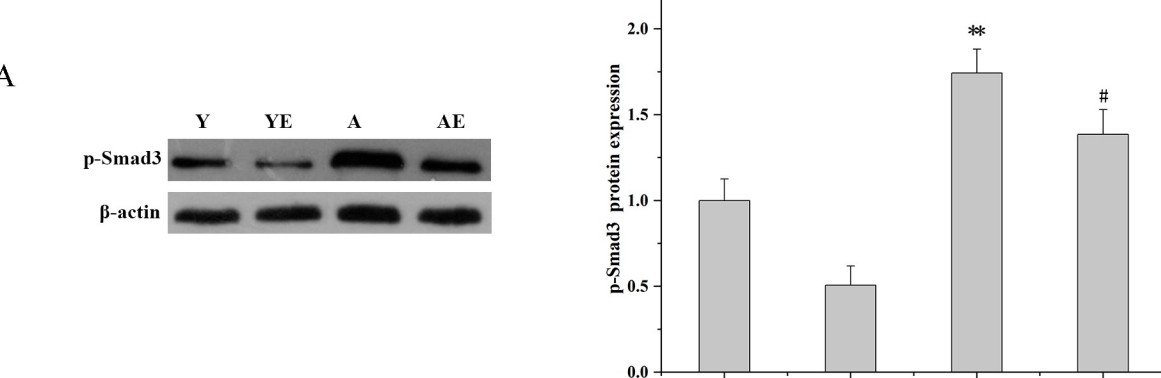

**Fig 5.** Aerobic exercise might suppress the expression of the TGF-β1/Smad3 signaling pathway (A) Western blot analysis (left) and quantification (right) of intrarenal protein levels of p-Smad3.*$P<0.05$ vs. Y; #$P<0.05$ vs. A; **$P<0.01$ vs. Y; ##$P<0.01$ vs. A. Data are expressed as mean±standard error (n = 6). Abbreviations: A.U. Arbitrary units; Y: young group; YE: young exercise group; A: aging group; AE: aging exercise group.

compared to the siRNA-control group, and significantly lower in the siRNA-Klotho+Ex group compared to the siRNA-Klotho group (Fig 6B). This result suggested that aerobic exercise ameliorated the renal injury associated with the inhibition of Klotho expression. Compared with the siRNA-control group, E-cadherin expression was significantly decreased, while that of α-SMA and Fibronectin were significantly increased in the siRNA-Klotho group. Further, the renal fibrosis indexes were significantly improved in the mice subjected to aerobic exercise intervention (Fig 6C and 6D).

### Role of aerobic exercise on upstream and downstream pathways in Klotho-inhibited mice

It has been shown in the paper that miR-34a regulates Klotho expression as the upstream of Klotho, therefore, miR-34a mRNA was tested with its upstream p-P53 protein expression. and unexpectedly, we found that miR-34a and p-P53 expression were significantly elevated in mice in the siRNA-Klotho group, compared with the siRNA-control group, and a more reasonable explanation for this phenomenon is given in the Discussion. MiR-34a and p-P53 were significantly decreased after exercise ($P<0.05$)(Fig 7A, 7B). Finally, we also examined the expression of Klotho's target genes and their downstreams, and the mRNA expression of these genes was down-regulated after exercise compared to the siRNA-Klotho group ($P<0.05$)(Fig 7A and 7C). These findings were also validated by Western blot analysis. (Fig 7B and 7D). The above findings suggest that aerobic exercise alleviates renal fibrosis by inhibiting the upstream pathway of Klotho, TGFβ1/p53/miR-34a, upregulating Klotho expression, followed by the inhibition of the activation of its downstream pathways, Wnt/β-catenin/MMP7 and TGFβ1/Smad3, thereby ameliorating renal fibrosis.

### Discussion

In recent years, renal fibrosis has emerged as a prevalent condition among the elderly population. Yet, the intricate molecular mechanisms governing the development of renal fibrosis are not completely understood, and effective strategies to mitigate its development are lacking. In this study, we postulated that aerobic exercise may up-regulate Klotho expression to ameliorate age-associated renal fibrosis. The results showed that exercise inhibited the upstream TGF-β1/

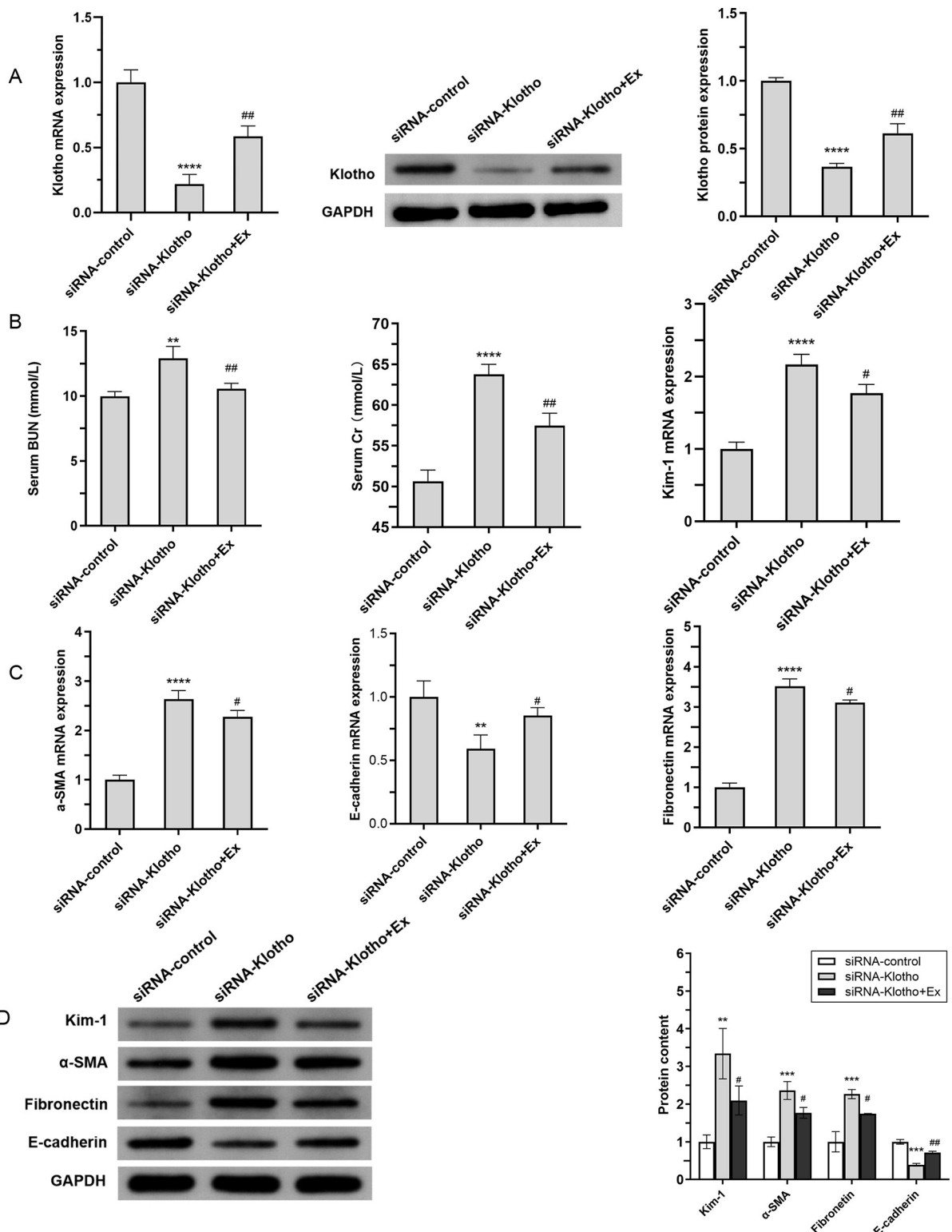

**Fig 6.** Rescue effect of aerobic exercise in Klotho-inhibited mice (A) The mRNA expression and protein levels of Klotho in the kidneys of mice in each group were analyzed by Q-PCR and Western blotting.(B) Serum BUN and Cr levels were assessed, and Kim-1 mRNA expression was detected by Q-PCR in the kidneys of each group of mice.(C)The mRNA expression of renal E-cadherin, a-SMA and Fibronectin in each group of mice was determined by Q-PCR. (D)Western blot analysis (left) and quantification (right) of intrarenal protein levels of Kim-1, E-cadherin, a-SMA, and Fibronectin. *$P<0.05$ vs. siRNA-control; #$P<0.05$ vs. siRNA-Klotho; **$P<0.01$ vs. siRNA-control;

##*P<0.01 vs. siRNA-Klotho; ***P<0.001 vs. siRNA-control; ###*P<0.001 vs. siRNA-Klotho; ****P<0.0001 vs. siRNA-control; ####*P<0.0001 vs. siRNA-Klotho; Data are expressed as mean±standard error (n = 6).

p53/miR-34a axis, to further suppress the activation of downstream TGF-β1/Smad3 and Wnt/β-catenin signaling pathways.

Klotho is considered a negative regulator of renal fibrosis, with its overexpression and exogenous administration found to inhibit renal fibrosis development [23]. For instance, administration of Klotho-derived peptide 1 via exogenous injection enhanced renal function, suppressed pro-fibrotic signaling pathways, restored endogenous Klotho expression, and mitigated renal fibrosis [6]. Furthermore, in murine models of unilateral ureteral obstruction, treatment with the activated triterpenoid acetyl-11-keto-β-boswellic acid derived from seahorse extract has been demonstrated to shield the kidneys from interstitial fibrosis by upregulating Klotho expression and suppressing pro-fibrotic signaling cascades [24]. Other studies have demonstrated that exercise can promote Klotho expression. For instance, a single session of acute resistance exercise in 12 untrained men elicited a significant increase in serum Klotho levels [25], while 12 weeks of moderate-intensity aerobic training in 11 healthy menopausal women led to heightened plasma Klotho concentrations and enhanced carotid stiffness [26]. Hence, in this study involving 19-month-old mice subjected to 12 weeks of aerobic exercise, we observed a significant increase in Klotho expression within renal tissue, concomitant with improved renal fibrosis associated with aging.

MiRNAs play important roles in the development and progression of renal fibrosis. For instance, a study by LU et al. demonstrated that elevated levels of miR103a-3p correlated with angiotensin II-induced renal injury and fibrosis. Elevated miR-103a-3p expression, coupled with decreased serine/threonine protein kinase levels in glomerular endothelial cells, results in excessive nuclear factor-κB/p65 activation, thereby fostering renal inflammation and fibrosis [27]. Similarly, another study found a significant increase in miR-34a expression in patients with renal fibrosis and in mice with induced infarction (UUO). Elevated levels of miR-34a lead to the downregulation of Klotho expression by directly targeting its 3'UTR, a process triggered by TGF-β1-mediated activation of p53. Dihydromyricetin administration effectively counteracted the aberrant elevation of miR-34a expression and the reduction of Klotho in UUO kidneys, resulting in notable amelioration of renal fibrosis [9]. Notably, the literature lacks reports on the impact of exercise on renal fibrosis through modulation of miRNA expression. Our experiments corroborated that aging precipitates a significant increase in TGF-β1/p53/miR-34a expression in renal fibrotic tissues. However, aerobic exercise intervention reversed the upregulation of miR-34a and the reduction of Klotho, thereby mitigating age-related renal fibrosis via inhibiting TGF-β1-induced p53 activation.

The Wnt/β-catenin pathway is the canonical route through which Wnt exerts its biological effects. Under physiological conditions in the adult kidney, the Wnt/β-catenin activity remains relatively quiescent. Cytoplasmic β-catenin is phosphorylated by the APC-Axin-GSK-3β complex, leading to its degradation and thereby maintaining Wnt signaling in an inhibited state. However, in renal fibrosis, Wnt molecules bind to cell membrane surface receptors, disrupting the formation of the APC-Axin-GSK-3β complex. Consequently, β-catenin is spared from phosphorylation and degradation, leading to its accumulation in the cytoplasm and subsequent translocation into the nucleus. In the nucleus, β-catenin interacts with the transcription factor TCF/LEF, thereby upregulating the transcription and expression of downstream target genes such as matrix metalloproteinase 7 (MMP7). Elevated MMP levels can disrupt the tubular basement membrane, worsening tubular epithelial cell damage and ultimately contributing to renal fibrosis. [28]. Klotho, which act an endogenous inhibitor of the Wnt pathway, forms a

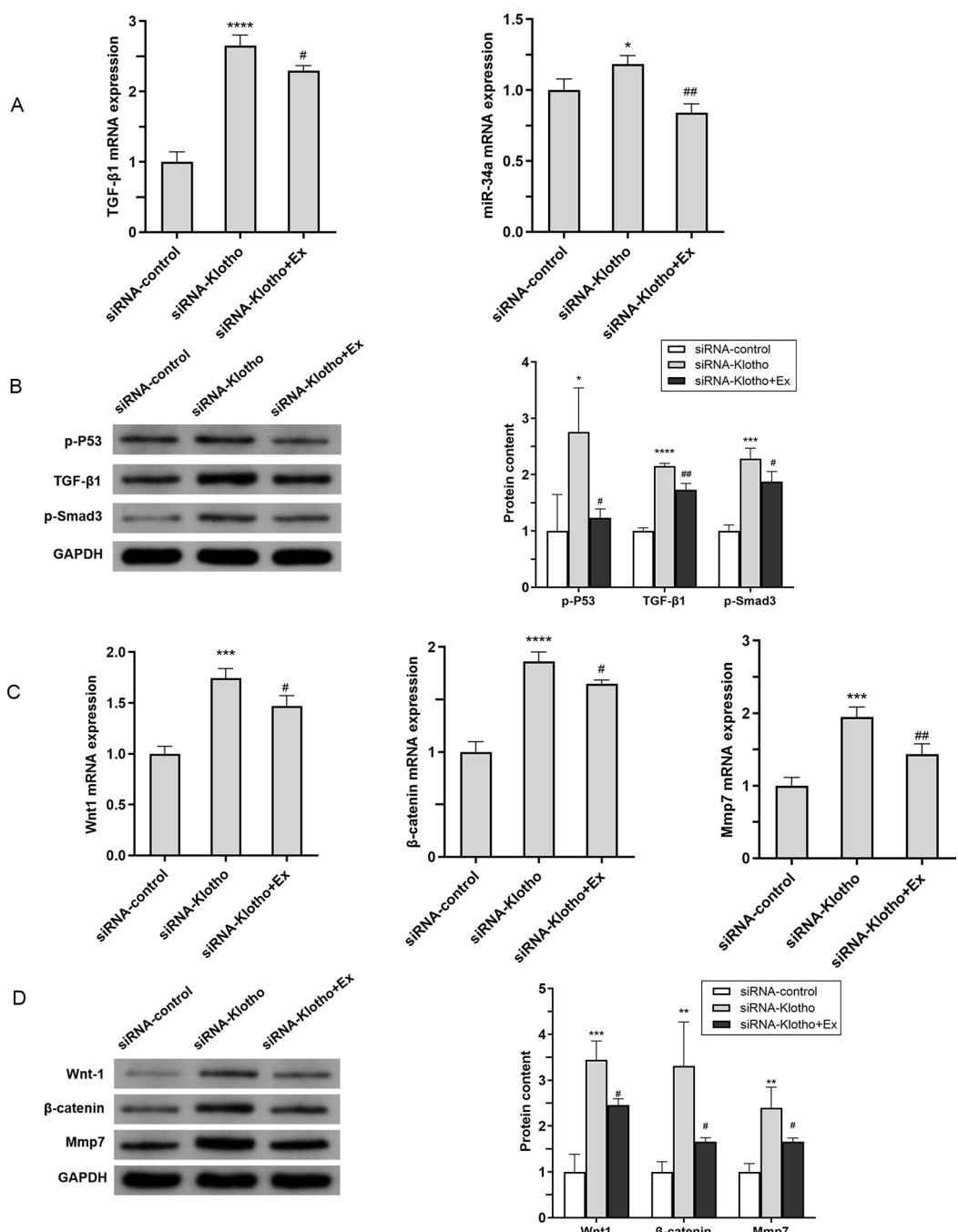

**Fig 7.** Role of aerobic exercise on upstream and downstream pathways in Klotho-inhibited mice (A-B)In the upstream and downstream signaling pathways of Klotho involving TGF-β1/p53/miR34a and TGF-β1/Smad3, renal mRNA expression of TGF-β1 and miR-34a was assessed using qPCR, while renal protein levels of TGF-β1, p-P53, and p-Smad3 were analyzed via Western blot.(C) The mRNA expression of Wnt1, β-catenin and Mmp7 in the kidney of each group of mice was determined by Q-PCR. (D) Western blot analysis (left) and quantification (right) of intrarenal protein levels of Wnt1, β-catenin and Mmp7. *$P$<0.05 vs. siRNA-control; #$P$<0.05 vs. siRNA-Klotho; **$P$<0.01 vs. siRNA-control; ##$P$<0.01 vs. siRNA-Klotho; ***$P$<0.001 vs. siRNA-control; ###$P$<0.001 vs. siRNA-Klotho; ****$P$<0.0001 vs. siRNA-control; ####$P$<0.0001 vs. siRNA-Klotho; Data are expressed as mean±standard error (n = 6).

complex with Wnt1, 4, and 7a, thereby impeding the binding of Wnt proteins to their receptors. This facilitates the formation of the APC-Axin-GSK-3β complex, leading to β-catenin phosphorylation and subsequent degradation, thereby deactivating the Wnt/β-catenin pathway and conferring a renal protective effect [29]. Miao et al. demonstrated that Wnt/β-catenin and RAS activity are upregulated in the aging kidney, while Klotho, acting as an antagonist of endogenous Wnt/β-catenin activity, attenuates d-galactose-induced renal fibrosis in a mouse model of accelerated aging, safeguarding renal mitochondrial function by curtailing reactive oxygen species production [30]. Chen et al. demonstrated that aerobic exercise can mitigate pulmonary fibrosis by increasing Klotho and FGF23 expression and inhibiting the Wnt1/β-catenin signaling pathway in a bleomycin-induced mouse model. [31]. In our experiments, we provide evidence suggesting that aerobic exercise may delay age-related renal fibrosis by enhancing Klotho expression and suppressing the Wnt1/β-catenin signaling pathway.

TGF-β1 is widely acknowledged as the principal driver of organ fibrosis and holds a pivotal role in the initiation of renal fibrosis [32]. In fibrotic conditions, the TGF-β1/Smad3 signaling is activated, and the Klotho protein competes with TGF-β1 for binding to the type II receptor of TGF-β1. This competition leads to a reduction in the binding of TGF-β1 to cell surface receptors in response to increased Klotho expression. A recent investigation demonstrated that a Klotho-derived peptide 1 in combating renal fibrosis by binding to TβR2, thereby disrupting the interaction between TGF-β and TβR2, suppressing fibroblast activation, and thwarting TGF-β-induced Smad2/3 activation [6]. Nevertheless, the role of exercise in ameliorating renal fibrosis via modulation of Klotho expression remains underexplored. Our findings suggest that aerobic exercise can inhibit the TGF-β1/Smad3 signaling pathway by modulating Klotho expression, thereby mitigate age-related renal fibrosis.

Previous studies have supported the concept that Klotho acts as a downstream target of miR-34a, whereby heightened miR-34a levels in the kidney contribute to reduced Klotho expression, thereby facilitating the activation of pro-fibrotic signaling pathways [15]. Hence, our study aimed to downregulate Klotho and determine whether aerobic exercise could directly enhance Klotho expression and its downstream targets. We found that Klotho inhibition not only promoted the activation of its downstream pro-renal fibrosis pathways (TGF-β1/Smad3, Wnt1/β-catenin) but also elevated its upstream expression of TGF-β1/p-P53/miR-34a. Aerobic exercise significantly increased Klotho expression while simultaneously suppressing its associated upstream and downstream signaling pathways. Consequently, a disease-perpetuating cycle mediated by TGF-β1 activation and Klotho reduction may participate in this process. Following Klotho inhibition, the TGF-β1/Smad3 expression was significantly upregulated, exacerbating renal fibrosis, and stimulating TGF-β1 subsequently activated the TGF-β1/p53/miR-34 signaling pathway upstream of Klotho. This cascade led to a decline in Klotho expression, further amplifying TGF-β1 expression and promoting the progression of renal fibrosis, thereby engendering a positive feedback loop consistent with the pathological cycle of the disease. Aerobic exercise has the potential to disrupt this deleterious cycle, fostering a favorable feedback loop that ameliorates renal fibrosis, thereby serving as a crucial strategy for delaying its onset(Fig 8A). Furthermore, our findings provide a theoretical foundation supporting the utilization of aerobic exercise to mitigate the development of renal fibrosis in aging.

Previous research has demonstrated the potential of exercise to ameliorate renal fibrosis [20, 21]. However, it is noteworthy that a singular exercise intensity may lead to intolerant outcomes in aging mice or may be ineffectiveness if the intensity is insufficient. Therefore, in elderly patients with CKD and concurrent comorbidities, it is crucial to ensure the safety and efficacy of the exercise regimen. Hence, future experiments should focus on developing personalized exercise prescriptions to optimize safety and efficacy.

A

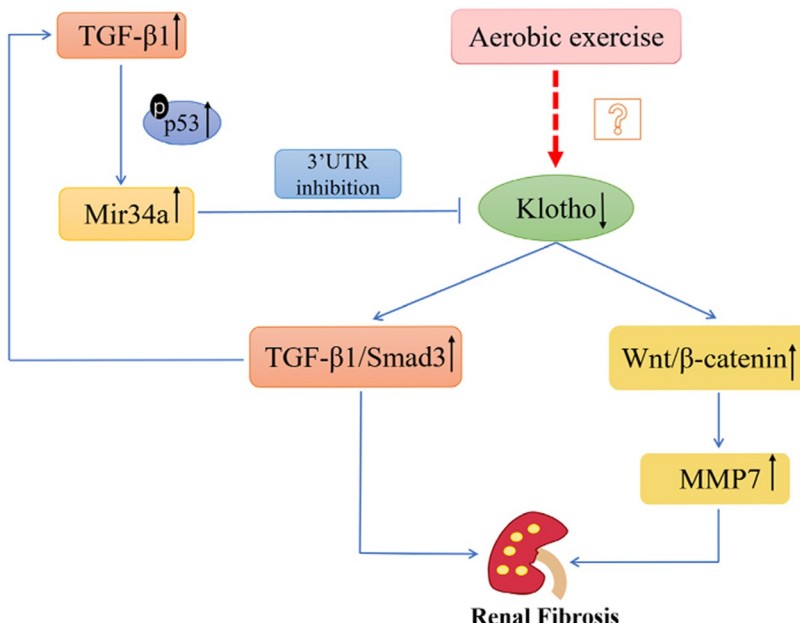

**Fig 8. Disease-perpetuating cycle mediated by TGF-β1 activation and Klotho reduction may be implicated in this process.** Aerobic exercise has the potential to disrupt this deleterious cycle.Aerobic exercise may increase the expression of Klotho by inhibiting the activation of the pro-fibrotic signaling pathway TGF-β1/p53/mir34a, thereby inhibiting the activation of the pro-fibrotic TGF-β1/smad3 and Wnt/β-catenin signaling pathways. Thereby effectively ameliorating aging-related renal fibrosis.

This study adds new insights into the mounting evidence of benefits from aerobic training to prevent and treat aging on renal fibrosis. Besides, exercise training could exert beneficial effects on blood pressure, lipids, glucose tolerance, bone density, depression, and quality of life [33]. These evidences suggest that exercise is an important strategy for active aging, which encourages more elderly participate in physical activities. In addition, Klotho may be an attractive new therapy to mimic the ameliorative effects of aerobic exercise on ageing-induced renal fibrosis. AS many patients with renal fibrosis have other age-related conditions or comorbidities (for example, arthritis and other musculoskeletal disorders) that preclude them from engaging in regular physical exercise. Therefore, Bolstering Klotho levels, either pharmacologically or through exercise, may thus constitute a novel therapeutic strategy to protect and/or delay the progression of renal fibrosis.

In conclusion, our study underscores the significance of aberrant Klotho expression in driving renal fibrosis progression, while highlighting the potential of aerobic exercise to enhance Klotho expression by attenuating the activation of the TGF-β1/p53/miR34a signaling pathway. This inhibition consequently mitigates the activation of the TGF-β1/Smad3 and Wnt/β-catenin signaling pathways, establishing a delayed and benign feedback loop in renal fibrosis. These findings offer novel insights that can be leveraged to develop prevention and treatments forrenal fibrosis.

## Supporting information

**S1 Raw image. WB three original figures.**
(PDF)

**S1 File.**
(DOCX)

**S2 File.**
(XLSX)

## Acknowledgments

The authors thank the Institute of Biological Sciences Shanxi University for professional technical assistance and the China Institute of Radiation Protection for animal care.

## Author Contributions

**Conceptualization:** Jinfeng Zhao.

**Data curation:** Yue Guan, Yinghua Chen, Yue Cai.

**Funding acquisition:** Jinfeng Zhao, Yue Cai.

**Methodology:** Jinfeng Zhao, Yue Guan, Yingxiu Jia.

**Validation:** Yue Cai.

**Writing – original draft:** Yingxiu Jia.

**Writing – review & editing:** Yue Guan, Yinghua Chen.

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
