## [Decision Letter · Decision Letter 0]

11 Jul 2024

PONE-D-24-16303Aerobic Exercise Up-regulates Klotho to Improve Renal Fibrosis Associated with Aging and its MechanismPLOS ONE

Dear Dr. CAI,

Thank you for submitting your manuscript to PLOS ONE. After careful consideration, we feel that it has merit but does not fully meet PLOS ONE’s publication criteria as it currently stands. Therefore, we invite you to submit a revised version of the manuscript that addresses the points raised during the review process.

**ACADEMIC EDITOR: **There are many questions about the completeness of this manuscript. Nevertheless, the reviewers provided many comments on possible improvements to this manuscript. The manuscript must be revised to reflect this. **- **Please format the manuscript according to the manuscript writing guidelines of PLoS One. Please ensure compliance with basic manuscript writing methods (e.g., spacing, capitalization, italics, special characters).

We look forward to receiving your revised manuscript.

Kind regards,

Kyung-Wan Baek, Ph.D.

Academic Editor

PLOS ONE

Journal Requirements:

5. Please update your submission to use the PLOS LaTeX template. The template and more information on our requirements for LaTeX submissions can be found at http://journals.plos.org/plosone/s/latex.

6. Please upload a copy of Supporting Information Figure/Table/etc. "Supplementary Materials1"  which you refer to in your text on page 6.

**Additional Editor Comments:**

Reviewers pointed out multiple areas in need of improvement in this manuscript, indicating the need for refinement.

Reviewers' comments:

Reviewer's Responses to Questions

**Comments to the Author**

1. Is the manuscript technically sound, and do the data support the conclusions?

Reviewer #1: Yes

Reviewer #2: Yes

2. Has the statistical analysis been performed appropriately and rigorously? 

Reviewer #1: Yes

Reviewer #2: Yes

3. Have the authors made all data underlying the findings in their manuscript fully available?

Reviewer #1: Yes

Reviewer #2: Yes

4. Is the manuscript presented in an intelligible fashion and written in standard English?

Reviewer #1: Yes

Reviewer #2: No

5. Review Comments to the Author

Reviewer #1: This article explores the effects and mechanisms of aerobic exercise on aging-related renal fibrosis through detailed experimental design and data analysis. The study results indicate that aerobic exercise can significantly improve ageing-related renal impairment by regulating Klotho expression and its related signalling pathways. While the article is well-structured and the data is comprehensive, there is still room for improvement in the experimental details, figure clarity, and consistency of statistical analysis.

1. The introduction section provides a comprehensive review of existing research, but the novelty and specific hypotheses of this study could be further highlighted.

2. The clarity of the figures is too low, making it difficult to read the text in the figures.

3. In the results section, some 'p' values are in lowercase and italicised, while others are uppercase and not italicised. Please ensure consistency throughout the manuscript. The results section should include more specific data comparisons and statistical significance to ensure the reproducibility and reliability of the data.

4. It is recommended to include a discussion of future research directions and the potential clinical implications of the findings.

5. P11, lines 212-214. The statement "BUN, Cr, and Kim-1 were significantly increased in the A group compared to the young group and improved in the AE group compared to the A group, indicating that aerobic exercise improved aging-related renal impairment" should be supported by quantitative data. The figures provided do not clearly show significant increases. Please include quantitative descriptions.

6. P6, lines 115-128. The steps for establishing the Klotho knockdown model are too brief. More detailed experimental steps and conditions should be provided.

7. The description of the statistical analysis for multiple group comparisons is not detailed enough. The specific comparisons between groups and their statistical significance should be clearly stated.

Reviewer #2: 1.An extensive editing for language and style is needed. A professional editing may be needed. A certificate for such an editing and language quality is needed when the revised manuscript is submitted.

2. The authors need to supplement the repeated results of three experiments and the WB results need to be co-displayed with β-actin or GAPDH on a PVDF membrane.

6. PLOS authors have the option to publish the peer review history of their article (what does this mean?). If published, this will include your full peer review and any attached files.

Reviewer #1: **Yes: **Wenfei Zhu

Reviewer #2: **Yes: **Yongfang Gong

---

## [Author Response · Author response to Decision Letter 0]

23 Aug 2024

Dear Editor and Reviewers:

Thank you for your careful review and positive comments on our manuscript. We have carefully considered your comments and revised them carefully. In the revised original manuscript, all changes made to the text are highlighted in red, and language touch-up modifications are highlighted in blue for easier identification; we hope that these changes will improve our manuscript. The following are our answers to the questions posed by the editor and reviewers.

Sincerely,

Corresponding author.

Response to Editor:

Q1: Please ensure that your manuscript meets PLOS ONE's style requirements, including those for file naming. The PLOS ONE style templates can be found at 

Answer 1: We sincerely appreciate your valuable comments. We have scrutinized the literature and revised it according to the style requirements of PLOS ONE, and the file naming has also been checked and revised.

Q2: We note that the grant information you provided in the ‘Funding Information’ and ‘Financial Disclosure’ sections do not match. 

Answer 2: Thank you for your reminder. We apologize for our careless error. When we resubmit, we will further provide and verify the correct grant number for the award received for research in the ‘Funding Information’ section.

Q3: We note that your Data Availability Statement is currently as follows: All relevant data are within the manuscript and its Supporting Information files.

Please confirm at this time whether or not your submission contains all raw data required to replicate the results of your study. Authors must share the “minimal data set” for their submission. PLOS defines the minimal data set to consist of the data required to replicate all study findings reported in the article, as well as related metadata and methods.

Answer 3:Thank you again for your careful review, we apologize for not uploading the raw data for the study in the Supporting Information file when we first submitted it, we will upload this data in the Supporting Information as you suggested. We have provided the mean, standard deviation, p-value, and raw values used to construct the graphs after processing the data in an excel file named “S3 original data”.

Q4: PLOS ONE now requires that authors provide the original uncropped and unadjusted images underlying all blot or gel results reported in a submission’s figures or Supporting Information files. This policy and the journal’s other requirements for blot/gel reporting and figure preparation are described in detail at https://journals.plos.org/plosone/s/figures#loc-blot-and-gel-reporting-requirements and https://journals.plos.org/plosone/s/figures#loc-preparing-figures-from-image-files. When you submit your revised manuscript, please ensure that your figures adhere fully to these guidelines and provide the original underlying images for all blot or gel data reported in your submission. See the following link for instructions on providing the original image data: https://journals.plos.org/plosone/s/figures#loc-original-images-for-blots-and-gels.   

Answer 4:Thank you for your careful review and positive evaluation of our manuscript. We have carefully considered your opinion, providing the raw, uncropped, and unadjusted images in the Supporting Information S1_raw_images. It is worth noting that we apologize that no single target protein on the entire membrane was detected in our initial Western blot experiments. Based on the consideration of saving experimental time and cost, we cut the bands containing the target proteins according to molecular weight after transfer and incubated them with antibodies separately. But in the subsequent experiments, we performed the experiments on the whole membrane, and images of the whole membrane have been included in the Supporting Information S1_raw_images.

Q5: Please update your submission to use the PLOS LaTeX template. The template and more information on our requirements for LaTeX submissions can be found at http://journals.plos.org/plosone/s/latex.

Answer 5:Thank you for your comments. We have carefully referenced the LaTeX submission requirements template in this submission and have modified the manuscript accordingly.

Q6: Please upload a copy of Supporting Information Figure/Table/etc. "Supplementary Materials1"  which you refer to in your text on page 6.

Answer 6:We sincerely thank you for your careful reading and apologize for our careless mistake. As per your suggestion, we have submitted this part of the support information in a supplementary document named “S2 supplement materials1”.

Responses to Reviewers：

Reviewer #1:This article explores the effects and mechanisms of aerobic exercise on aging-related renal fibrosis through detailed experimental design and data analysis. The study results indicate that aerobic exercise can significantly improve ageing-related renal impairment by regulating Klotho expression and its related signalling pathways. While the article is well-structured and the data is comprehensive, there is still room for improvement in the experimental details, figure clarity, and consistency of statistical analysis.

Q1: The introduction section provides a comprehensive review of existing research, but the novelty and specific hypotheses of this study could be further highlighted.

Answer 1:We thank you for your positive and constructive comments on our paper. After much deliberation, we have further summarized the introduction section of the article to add to the novelty and specific hypotheses of our study, and we have marked the revisions in red in lines 83-87 of the text.

Q2: The clarity of the figures is too low, making it difficult to read the text in the figures.

Answer 2: We appreciate your feedback, which will help improve the quality of our paper. In order to make the images clearer, we have adjusted the figures according to the requirements, such as adjusting the size and clarity of the figures, etc. We have resubmitted the modified figures, and we hope that our adjusted and resubmitted figures will be up to the standard of the journal.

Q3: In the results section, some 'p' values are in lowercase and italicised, while others are uppercase and not italicised. Please ensure consistency throughout the manuscript. The results section should include more specific data comparisons and statistical significance to ensure the reproducibility and reliability of the data.

Answer 3: We sincerely thank the reviewers for their careful reading and apologize for our careless mistakes. According to your suggestion, we have changed ‘P’ to a uniform capitalized and italicized form. In the results section, we have also indicated more specific data comparisons and statistical significance, for example, in lines 218-223: ‘Moreover, the expression level of the tubular epithelial cell marker E-calmodulin (E-cadherin) was significantly decreased in the renal tissue of group A, and that of the renal interstitial markers a-SMA and fibronectin was significantly increased, indicating that aging was accompanied by renal fibrosis. Notably, the aging-associated changes were improved in the AE group following the 12 weeks of aerobic exercise intervention (Fig 1D-E).’ has been changed to ‘Moreover, the expression level of the tubular epithelial cell marker E-calmodulin (E-cadherin) was significantly decreased in the renal tissue of group A (P<0.05), and that of the renal interstitial markers a-SMA and fibronectin was significantly increased (P<0.01), indicating that aging was accompanied by renal fibrosis. Notably, the aging-associated changes were improved in the AE group following the 12 weeks of aerobic exercise intervention (P<0.05) (Fig 1D-E). ’Others have been highlighted in red in the “Revised Manuscript with Track Changes” document.

Q4 :It is recommended to include a discussion of future research directions and the potential clinical implications of the findings.

Answer 4: Thank you for your expert advice. We have revised the discussion of future research directions and potential clinical implications of the findings in lines 484-495 of the text in red.

Q5 :P11, lines 212-214. The statement "BUN, Cr, and Kim-1 were significantly increased in the A group compared to the young group and improved in the AE group compared to the A group, indicating that aerobic exercise improved aging-related renal impairment" should be supported by quantitative data. The figures provided do not clearly show significant increases. Please include quantitative descriptions.

Answer 5: We sincerely thank you for your careful reading. According to your suggestion, we have also made changes to the quantitative data support in the results, which are marked in red in lines 206-213, and we have also provided specific quantitative data for BUN and Cr in Table. 3 in the supplementary file of “S2 supplement materials1”, and thank you again for your reasonable suggestions.

Q6 :P6, lines 115-128. The steps for establishing the Klotho knockdown model are too brief. More detailed experimental steps and conditions should be provided.

Answer 6: Thank you for your suggestion. In order to make our experimental description more detailed, we have described the steps of the Klotho knockdown model in more detail in lines 118-135 in the experimental methodology, and we have marked the changes in red.

Q7 :The description of the statistical analysis for multiple group comparisons is not detailed enough. The specific comparisons between groups and their statistical significance should be clearly stated.

Answer 7: Thank you very much for your suggestion, firstly, we have made additions to the statistical analysis in lines 197-201 of the Methods section, and secondly, we have added the P-value descriptions for the specific comparisons between groups in some of the results, which have been marked in red in the text, and we are willing to continue to make modifications in accordance with your comments if they are not perfect enough.

Reviewer #2: 

Q1 :An extensive editing for language and style is needed. A professional editing may be needed. A certificate for such an editing and language quality is needed when the revised manuscript is submitted.

Answer 1: Thank you for your suggestions. We did our best to improve the manuscript and made some changes to it. And we did invite a native English speaker from the United States to help touch up our article. A certificate of editorial language quality is provided here. These changes we made do not affect the content or framework of the paper. Instead of listing the changes here, we have highlighted them in blue in the revised paper. We sincerely thank the editors and reviewers for their enthusiastic work and hope that this revision will be approved.

Q2 :The authors need to supplement the repeated results of three experiments and the WB results need to be co-displayed with β-actin or GAPDH on a PVDF membrane.

Answer 2:Thank you for your careful review and positive evaluation of our manuscript. We have carefully considered your opinion, we complement the replicate results of the three experiments of the WB experiment in the Supporting Information S1_raw_images. It is worth noting that we apologize that no single target protein on the entire membrane was detected in our initial Western blot experiments. Based on the consideration of saving experimental time and cost, we cut the bands containing the target proteins according to molecular weight after transfer and incubated them with antibodies separately. But in the subsequent experiments, we performed the experiments on the whole membrane, and images of the whole membrane have been included in the supplementary materials S1_raw_images.

---

## [Decision Letter · Decision Letter 1]

12 Sep 2024

Aerobic exercise up-regulates Klotho to improve renal fibrosis associated with aging and its mechanism

PONE-D-24-16303R1

Dear Dr. Cai,

We’re pleased to inform you that your manuscript has been judged scientifically suitable for publication and will be formally accepted for publication once it meets all outstanding technical requirements.

Kind regards,

Kyung-Wan Baek, Ph.D.

Academic Editor

PLOS ONE

Additional Editor Comments (optional):

One of the reviewers for this manuscript has repeatedly declined to conduct further reviews. However, the reviewer initially recommended minor revisions, which have been confirmed as completed. Therefore, we have decided to accept the manuscript without any additional review.

Reviewers' comments:

Reviewer's Responses to Questions

**Comments to the Author**

1. If the authors have adequately addressed your comments raised in a previous round of review and you feel that this manuscript is now acceptable for publication, you may indicate that here to bypass the “Comments to the Author” section, enter your conflict of interest statement in the “Confidential to Editor” section, and submit your "Accept" recommendation.

Reviewer #1: All comments have been addressed

2. Is the manuscript technically sound, and do the data support the conclusions?

Reviewer #1: Yes

3. Has the statistical analysis been performed appropriately and rigorously? 

Reviewer #1: Yes

4. Have the authors made all data underlying the findings in their manuscript fully available?

Reviewer #1: Yes

5. Is the manuscript presented in an intelligible fashion and written in standard English?

Reviewer #1: Yes

6. Review Comments to the Author

Reviewer #1: I have no further comments. All the comments have been addressed. This study has focused on an important issue in this area.

7. PLOS authors have the option to publish the peer review history of their article (what does this mean?). If published, this will include your full peer review and any attached files.

Reviewer #1: No

---

## [Editor Report · Acceptance letter]

17 Sep 2024

PONE-D-24-16303R1 

PLOS ONE

Dear Dr. Cai, 

I'm pleased to inform you that your manuscript has been deemed suitable for publication in PLOS ONE. Congratulations! Your manuscript is now being handed over to our production team.

Kind regards, 

on behalf of

Dr. Kyung-Wan Baek 

Academic Editor

PLOS ONE